# Indices of Cardiovascular Health, Body Composition and Aerobic Endurance in Young Women; Differential Effects of Two Endurance-Based Training Modalities

**DOI:** 10.3390/healthcare9040449

**Published:** 2021-04-11

**Authors:** Kemal Idrizovic, Gentiana Beqa Ahmeti, Damir Sekulic, Ante Zevrnja, Ljerka Ostojic, Sime Versic, Natasa Zenic

**Affiliations:** 1Faculty for Sport and Physical Education, University of Montenegro, 81400 Niksic, Montenegro; kemo@t-com.me; 2Faculty of Physical Education and Sport, University of Prishtina, 10000 Prishtina, Kosovo; gentiana.beqa@uni-pr.edu; 3Faculty of Kinesiology, University of Split, 21000 Split, Croatia; dado@kifst.hr (D.S.); ljerka.ostojic@mef.sum.ba (L.O.); sime.versic@kifst.hr (S.V.); 4Faculty of Medicine, University of Mostar, 88000 Mostar, Bosnia and Herzegovina; antezevrnja17@gmail.com; 5Clinical Hospital Split, 21000 Split, Croatia

**Keywords:** aerobic endurance, exercise, health indicators, aerobic dance, physical capacity

## Abstract

Endurance training (ET) has multiple beneficial effects on cardiovascular health (CVH), but there is an evident lack of knowledge on differential effects of various types of ET on indices of CVH in women. The aim of this study was to analyse the effectiveness of two different types of ET on changes in indicators of CVH in apparently healthy adult women. The sample included 58 women (24 ± 3 years; height: 165 ± 6 cm, mass: 66.7 ± 7.2 kg, BMI: 24.3 ± 2.5 kg/m^2^, at baseline) divided into one control non-exercising group (*n* = 19), and two exercising experimental groups (EE). The first EE participated in choreographed aerobic-endurance training (CAT; *n* = 19), while the second participated in treadmill-based endurance exercise (TEE; *n* = 20) during the experimental protocol (8 weeks, 24 training sessions). The testing included pre- and post-exercise protocols and measures of anthropometric/body composition indices, lipid panel, and endurance capacity. Two-way analysis of variance for repeated measurements with consecutive post hoc analysis was applied to the “group” and “measurement” variables. The main significant ANOVA effects found for measurement, and “Group x Measurement” interaction (*p* < 0.05) were found for all variables but body height. The EE induced positive changes in lipid panel variables, anthropometric/body-build status, and endurance capacity. However, TEE improved endurance capacity to a greater extent than CAT. The results suggest that that the optimal exercise intensity and self-chosen type of physical-activity may result in positive effects on indices of CVH, even in women of young age and good health status.

## 1. Introduction

Physical activity (PA) has a positive effect on various indices of health status, including the entire locomotor system, and quality of life [1]. It also helps with controlling body weight and has positive effects on psychological status [2]. Moreover, studies regularly confirm the positive preventive and therapeutic effects of increased PA on many chronic diseases, such as cardiovascular and tumour diseases [3,4]. Therefore, increase in PA is recognised as a worldwide public health priority [5,6,7]. Nowadays, increase in PA activity is most commonly associated with the inclusion of a certain type of physical exercise (PE) in the individuals’ daily rhythm. More specifically, while PA is a broader term and refers to any movement performed with the help of the skeletal muscles during which the energy expenditure exceeds the resting level, PE implies a planned and programmed PA performed with a specific goal [8].

There are many different variations of PE, but the most basic classification scheme includes resistance training, endurance training, and combined training [8]. Indeed, it is well known that different forms of exercise have different effects on overall health and anthropological status and, consequently, recommendations regarding the type, amount, intensity, and frequency of exercise vary [9]. Specifically, endurance training develops cardiovascular and respiratory functions the most, while resistance training primarily influences strength and power, and other muscular capacities [10,11]. Moreover, within the same type of PE there are certain differences in activity, so the impact of endurance training varies depending on the extent, intensity, and structure of movement [12].

One of the most important benefits of endurance training is the decrease in cardiovascular risk (e.g., probability of developing a cardiovascular disease), and improvement of cardiovascular health [13]. Although the cardiovascular risk is higher in men [14], low levels of PA in women today highlight the importance of PE in reducing cardiovascular risk factors in this group [15,16]. Among the many indicators of cardiovascular health, several variables deserve special attention, including variables of lipid panel, and anthropometric/body build status.

Lipid panel (i.e., lipid test, lipid profile) is a set of variables describing the lipid levels in the blood, including total cholesterol (TC), high-density lipoprotein cholesterol (HDL), low-density lipoprotein cholesterol (LDL), and triglycerides (TG) [17]. The strategies to reduce cardiovascular risk in primary and secondary prevention regularly focus on control (i.e., reduction) of lipid panel variables, particularly LDL [18]. Additionally, there is a global consensus that poor cardiovascular health is strongly correlated with overweight/obesity, with higher cardiovascular risk in overweight and obese individuals [8,19]. Therefore, evaluations of lipid panel, and anthropometric/body built status are valuable tools in assessing risk for cardiovascular problems [20,21]. Among other types of interventions, studies even have evaluated the effects of different types of PE on various indices of cardiovascular health status in women [8,16,22].

For example, Barranco-Ruiz et al. studied sedentary females and evidenced improvement in various cardiovascular risk variables as a result of 4-month long choreographed endurance PE (e.g., Zumba fitness program) [22]. Cebula et al. applied 6-week Nordic walking endurance exercise and evidenced significant improvement in cardiovascular function of sedentary postmenopausal women [23]. Chovanec and Gropel confirmed the positive effects of 8-week endurance-based training on cardiovascular stress responses in untrained healthy female students [24]. The effects of combined-type training (45 min of aerobic exercise and 20min of resistance training) in women were also studied, and positive effects on morphological measures, blood pressure, and lipid panel were found [15]. Finally, a recent study showed that both resistance training and endurance training have a positive impact on the cardiovascular health indices of healthy young women, with no significant differences between groups conducting different forms of PE [16].

Evidently, studies regularly confirmed the positive effects of endurance-based PE on cardiovascular health in women [16,23,24,25]. However, there is an obvious lack of studies which simultaneously examined the effects of different forms of endurance exercise on cardiovascular health indices in young adult women. This is especially important given the fact that PE programs are primarily chosen on the basis of personal preferences [26]. Additionally, motivation for PE is one of the most important predictors of persistence in PE and, consequently, it directly defines the efficiency of PE [27,28]. Moreover, personal preferences toward a certain type of PE are hardly modifiable, so it would be important to evaluate the effects of different forms of endurance exercise on indicators of cardiovascular health. This will provide us with a better understanding of the potential benefits of various types of PE, and therefore a more accurate prescription of the PE as a method of reducing the health risks associated with low PA.

The aim of this study was to evaluate the effectiveness of two modalities of endurance-exercise on changes in indicators of cardiovascular health in apparently healthy adult women. Specifically, we examined the effects of choreographed aerobic-endurance training (CAT) and treadmill-based endurance exercise (EET) on anthropometric/body-build indices and lipid profile variables in young adult females. We hypothesised that the studied PE programs will have a positive influence on the studied variables, with no differential effects regarding PE type.

## 2. Materials and Methods

### 2.1. Participants

Participants were 58 apparently healthy young women (24.1 ± 2.0 years; height: 166 ± 5.9 cm, mass: 66.9 ± 7.6 kg, BMI: 24.4 ± 2.7 kg/m^2^, at baseline). The participants were generally inexperienced in terms of PE, and all were members of one fitness centre in Prishtina, Kosovo. The participants were offered several types of PE, and based on their self-preferences they were divided into two exercising experimental groups (EE) and one age-matched non-exercising group (*n* = 19; 23.8 ± 2.0 years; height: 167 ± 6.1 cm, mass: 65.4 ± 8.3 kg, BMI: 23.8 ± 2.7 kg/m^2^, at baseline). The first EE performed choreographed aerobic-endurance training (CAT; *n* = 19; 24 ± 2 years; height: 166 ± 6.0 cm, mass: 67.9 ± 5.9 kg, BMI: 24.5 ± 2.5 kg/m^2^, at baseline). The second EE participated in treadmill-based endurance exercise (TEE; *n* = 20: 25 ± 2 years; height: 168.1 ± 7.2 cm, mass: 67.4 ± 8.4 kg, BMI: 25.1 ± 2.7 kg/m^2^, at baseline). Initially, the CAT group consisted of 23 participants, and 24 participants were involved in TEE program at the beginning. However, at the end of the study we included only those participants who participated in a minimum of 80% of training sessions, resulting in a drop-out rate of 17% (18% and 16% for CAT and TEE, respectively). The non-exercising group was formed of participants that showed interest in some of the observed types of PE but, at the moment of their arrival at the fitness centre, the selected program was unavailable due to limited space. The design of the study is presented in Figure 1

Regarding the participants’ health profiles, pre-treatment measurements showed that approximately 1/3 (32%) were overweight (BMI > 25 kg/m^2^), and every fourth had total cholesterol (TC) levels elevated above the optimal threshold [29]. All participants were informed about the aims and procedures of the study and agreed to participate by signing informed consent forms. The study was approved by the Ethical Board of the University of Split, Faculty of Kinesiology, Split, Croatia (EBO: 2141-6775-234).

### 2.2. Testing and Variables

Given the study design, testing occurred between two time points: before the treatment and 8 weeks after at the end of the PE protocols. The set of variables included (i) anthropometric body composition indices, (ii) lipid panel, and (iii) endurance capacity.

The lipid panel and anthropometric variables were measured in an accredited medical laboratory Biohem, (Gjakovë, Kosovo). Anthropometric/body indices included body height (BH), body mass (BM), and body mass index (BMI) derived from participants’ BH and BM, and three skinfolds on the triceps, thigh, and suprailiac. The skinfolds were used for the Jackson–Pollock and Siri formula to estimate body fat percentage (BF%) [30].

Body density = 1.0994921 − (0.0009929 × [triceps skinfold + thigh skinfold + suprailiac skinfold]) + (0.0000023 × [triceps skinfold + thigh skinfold + suprailiac skinfold] 2) − (0.0001392 × age),
BF% = (4.95/body density − 4.5) × 100

Anthropometric variables were measured according to the standard procedures of the International Society for the Advancement of Kinanthropometry [31]. A Seca stadiometer and scale (Seca, Birmingham, UK) and skinfold caliper (Holtain, London, UK) were used for measurements that occurred in the morning, before blood sampling. Blood samples were taken after overnight fasting in order to analyse plasma glucose (PG), TC, HDL, and TG. The LDL was calculated with The Friedwald equation [32]. Samples were collected in BD Vacutainer fi SSTII Advance vacuum tubes (BD, Plymouth, UK), centrifuged at 3500 rpm for 10 min (Centrifugal Hettich, Tuttlingen, Germany), and analysed using the COBAS Integra 400+ analyser (Roche Diagnostics International Ltd., Rotkreuz, Switzerland).

For the evaluation of endurance capacities, participants performed the Rockport Test on a treadmill. This test is used to indirectly estimate the level of maximal oxygen consumption (VO2max). The purpose of the test is to cover one mile via fast walking or running. Participants perform the test with heart rate chest belts and, after the test is finished, a VO2max score is calculated using the following equation for females [33]:VO2 = 139.168 − (0.388 × age) − (0.077 × weight in lb.) − (3.265 × walk time in minutes) −(0.156 × heart rate)

The testing of endurance capacities was organised in the same fitness centre in which the PE 8-week protocols occurred.

### 2.3. Physical Exercise Programs

PE programs occurred during an 8-week period, with three trainings per week and one day off between them. This time frame for PE interventions was selected as most drop-outs occur after two months of training [34]. Both PE programs had 24 sessions, and the inclusion criteria for participants was a minimum of 21 conducted training sessions. Training sessions were organised on Mondays, Wednesdays, and Fridays, between 16:00 and 21:00, and lasted from 45 min in the first weeks to 60 min at the end of the protocol, given the adaptation and improvement of the participants.

Nova 450 treadmills (Nova Sport, Istanbul, Turkey) were used for the TEE group, with speed ranging from 1 to 20 km/h and an incline between 0% and 15%. Prior to the first training sessions, all participants in the TEE group conducted a Conconi test. This test is used as an indirect method for evaluating the anaerobic threshold that can be seen as a deflection point in the linear association between running speed and heart rate [35]. This information allowed for the individualisation of training loads as participants wore heart rate chest belts every session and had to maintain their heart rates in the range of 5–30 beats below the anaerobic threshold. In order to maintain the desired thresholds, participants were allowed to change running speeds and incline based on the heart rate displayed on the treadmill monitor. Since adaptations of the cardiorespiratory and locomotor systems were expected due to training, participants had been introduced with continuous, interval, and fartlek running protocols and were given the instructions to choose one form of training per each week but were also left with the possibility of changing the running protocol during every training session. In general, all participants performed all types of running protocols, but always at an intensity below the anaerobic threshold. The intensity of exercise in the TEE groups was controlled during each training session using the heart rate monitors, and participants were instructed to keep the heart rate between 60% and 80% of the maximal heart rate (approximately 120–160 beats/min, depending on the participant’s characteristics). The Conconi test was repeated every 2 weeks in order to adequately correct the training load.

In the CAT group, participants performed aerobic dance routines. Generally, all CAT classes consisted of: (i) 5–10 min of warm-up (exercising at lower intensity, using the usual warm-up simple moves, and slower music tempo (120–145 beats per minute), followed by (ii) light dynamic pre-stretching (2–3 min), (ii) 30–40 min of the main part of the class (more choreographed movements performed at higher music pace of 140–180 bpm), and (iv) 5–15 min of cool down, and passive stretching. All participants, regardless of health and fitness status, performed the same protocol for the whole 8 weeks, which was controlled in real time by an experienced aerobic dance instructor who eventually increased/decreased the intensity of the exercise depending on perceived capacities of the group members. Specifically, at each training session three participants regularly wore heart rate monitors and controlled the intensity of exercise at each training session. They were instructed to report if their heart rate was below 120 beats/min, and/or raised above 160 beats/min. When two of three participants who wore a chest belt and controlled the training intensity reported the same discrepancy in heart rate (bellow or above the required frequencies), the instructor modified the training program either by increasing the exercise demands (i.e., by changing the amplitude of movement, adding/avoiding the jumps, increasing/decreasing the music tempo), which consequently resulted in increased/decreased intensity of the exercise. However, it cannot be said that the intensity of the CAT was individualised because the characteristics of the group-exercise program generally do not allow more accurate interventions in exercise intensity [34].

### 2.4. Statistics

The statistical analysis included (i) descriptive statistic parameters and (ii) methods for determining differences between groups and measurements.

Descriptive statistics included means and standard deviations, and Kolmogorov–Smirnov tests were used to evaluate the normality of the distributions for all observed variables. The homoscedasticity of the variables was checked with Levene’s test.

For the analyses of training effects, a two-way analysis of variance for repeated measurements (ANOVA) for the group (control—C, TEE and CAT) and measurement (pre- and post-treatment) variables was applied with consecutive Scheffe’s post hoc analysis. The partial eta squared values (η^2^) were also reported (small effect size (ES): >0.02; medium ES: >0.13; large ES: >0.26) as measures of effect sizes.

Statistica 13.5 (TIBCO Software Inc. Palo Alto, CA, USA) was used, and a significance level of *p* < 0.05 was applied for all calculations.

## 3. Results

Table 1 presents ANOVA results. In short, significant (*p* < 0.05) main effects with large ES for “Measurement” were evidenced for all study variables. Significant main effects for “Group” were found for PG (large ES), TC (small ES), TG (medium ES), and Rockport endurance capacity test (END) (medium ES). “Group x Measurement” interaction was significant for all variables with medium ES for PG, and large ES for the remaining variables.

Apart from significant decrease in BM for TEE and CAT, no significant post hoc differences were found for BM, and BMI. BF% decreased significantly in TEE and CAT, and both experimental groups had lower BF% than the C-group at post-test. All groups achieved better results in END at post-testing, with significant between-group differences when TEE was compared to C, and when TEE was compered to CAT (better results achieved by TEE in both cases). Significant within-group post hoc differences for both training groups were found for PG, TC, HDL, and LDL, with significant differences between TEE and C for post-testing (for all mentioned variables), and significant difference between CAT and C (for LDL only). TG decreased significantly in all three groups over the study course, with significant differences between TEE and C, and CAT and C in post-testing, with lower levels in training groups (Table 2).

Although the study did not include specific interventions in participants’ diet, nutrient intake, and caloric intake (in kcal) for all groups are evidenced before and at the end of the study period, and the results are presented in Figure 2. In brief, no significant differences between groups were evidenced.

## 4. Discussion

The main aim of this study was to evaluate the effectiveness of two different types of endurance training on various indices of cardiovascular health in healthy young women. Accordingly, there are several important findings. First, endurance-based exercise modalities induced positive changes in lipid panel and glucose variables. Moreover, both experimental groups progressed in their endurance capacities, while positive changes were shown in anthropometric/body-build status as well. Finally, participants who performed endurance-training on the treadmill improved their endurance capacity to a greater extent than participants who performed aerobic dance routines. Due to some differential effects of PE programs, our initial study hypothesis may be partially accepted.

### 4.1. Endurance Exercise, Lipid Panel, and Plasma Glucose Levels

With more than 7.5 million deaths per year, which amounts to 31% of all global deaths, cardiovascular diseases are the leading cause of death in developed countries [36,37]. Among others, high cholesterol concentrations and physical inactivity stand out as the most important factors contributing to the risk of cardiovascular death [38]. Studies have confirmed that even a small decrease in cholesterol levels directly reduces risks to cardiovascular health [36]. In addition to cholesterol-lowering substances, one of the most important factors in decreasing its values is regular and appropriate PE [8]. The results of our study are consistent with previous findings that regularly indicated a decrease in the value of total cholesterol (TC) and low-density lipoprotein cholesterol (LDL) as a result of regular PE [39]. Briefly, PE affects the production and activity of several enzymes that function to enhance the reverse cholesterol transport system, resulting in an increase in HDL and a decrease in LDL (the latter generally being the main source of artery-clogging plaque) [40].

Few studies have analysed the effects of PE on young, healthy women, but our results are generally in agreement with the results reported so far, which indicate positive effects of PE on cardiovascular health status, including lipid panel [41,42,43]. In particular, a study on a combined sample of healthy, inactive men and women showed significant reductions in total cholesterol, total cholesterol/HDL-C ratio, and diastolic blood pressure due to a 6-month intervention [41]. Reduced blood concentrations of triglycerides and significantly increased blood concentrations of HDL have been reported in young women after 16 weeks of aerobic training [44]. The results of our study indicate a positive influence of endurance-based exercise on reduction in plasma glucose (PG). These results are explainable by the knowledge that that carbohydrates, including PG, are the main source of energy in PE [45]. As key factors in the utilisation of PG, the authors highlight the intensity of exercise, which is reflected in the number of muscle fibres involved, and exercise extensity, as this partially compensates for the progressive decrease in muscle glycogen concentration [46].

Additionally, PE increases insulin sensitivity, which directly affects the reduction in PG [45,47]. In accordance with our results, previous studies confirm the influence of different types of PE on PG metabolism [48]. For example, a study in a combined sample of men and women found significant improvement in glucose metabolism and insulin sensitivity after a 4-week treatment that included combined endurance and resistance training [49]. Similar effects have been confirmed in different endurance training modalities like walking [50], running [51], cycling [52] and rowing [53].

One could argue that changes in PG levels could be a result of changes in nutritional habits as well. Indeed, changes in nutrition are known to be another influence on PG, even in young healthy people. However, it is not likely that nutrition significantly influenced our results, simply because our participants did not change their nutritional habits and caloric intake significantly (please see Results for details). On the other hand, each training session resulted in a caloric expenditure of 300–400 kcal. Therefore, we are of the opinion that changes in PG levels are mostly related to (i) the utilisation of the PG during exercise and (ii) increases insulin sensitivity.

### 4.2. Endurance Exercise, Anthropometric/Body-Built Indices and Endurance Capacity

Both groups that conducted endurance training intervention achieved positive changes in anthropometric/body-build indices. We believe these changes mostly result from (i) increased energy consumption and the resulting caloric deficit and (ii) the increased metabolism of endogenous energy stores (i.e., adipose tissue triglyceride) [54]. Generally, our results are consistent with previous findings, where aerobic activity has regularly been shown to be effective in inducing changes in anthropometric measures, especially in terms of subcutaneous adipose tissue reduction [25,42].

Specifically, in a study with sedentary but healthy women, the authors recorded positive changes in anthropometric/morphological parameters after 16 weeks of aerobic training [42]. Additionally, Spanish authors reported a significant loss of body fat after a 16-week intervention, which included aerobic-dance training [22]. One of the few studies in which no changes were found in anthropometric/body-build indices as a result of endurance training was on a sample of young women without weight problems (BMI below 26) [55]. In brief, despite six months of treatment, no significant decline in body fat and BMI was found, which was explained by the inability to control energy intake and a potential ceiling effect [55]. Specifically, Poelhman et al. stated that given the already low baseline levels, it was difficult to reduce total or visceral fat in young women [55]. Given the similarity between participants (healthy young women in both studies), a possible explanation for the differences between the obtained results (significant changes for anthropometric/body-build indices in our study vs. no changes in the investigation of Poelhman et al.) is provided next.

As presented in the Methods section, one of the intentions of our study was to investigate the effects of training programs that were personally chosen by each participant. In other words, our participants were not randomly allocated into the exercise programs, but selected PE based on their own preferences. Previous research that has analysed the training effects of PE randomly allocated the participants in the study groups, and this was done irrespective of the personal preferences towards some types of physical exercise [22,42,51,52]. One of the more important assumptions of the effectiveness of PE is regularity [26]. In order to achieve this, an individual must be interested in a particular type of training, and therefore they should choose it based on personal preferences [27,28]. We can therefore assume that the results achieved in our study, especially those related to changes in anthropometric/body-build indices, were at least partially caused via the personal choice of PE type, which contributed to participants’ motivation. One could argue that the changes we obtained can be partially a result of type of measurement, and questionable reliability of the method (i.e., we estimated BF% on the basis of skinfold measurement. However, the fact that all participants were measured by same evaluator, together with significant changes in training groups in comparison to non-significant changes of BF% in control group at least partially reduce the possibility that type of the measurement influenced our result.

Both PE programs improved aerobic endurance capacity. Similar findings were obtained by Finnish authors in a study on healthy middle-aged women [56]. The group that did aerobic endurance training on bicycles, with the load individualised as in our study, achieved an average VO2max increase of 23% after 21 weeks [56]. These changes are due to the adaptation to aerobic training and have been regularly reported elsewhere [57,58,59,60]. VO2max is influenced by the maximal cardiac output and the maximal arterial-mixed venous oxygen difference [58]. Aerobic training results in an increase in maximal stroke volume and skeletal muscle capillarisation, which consequently leads to an improvement in VO2max [57,61]. Therefore, although in our study the intervention was relatively short (i.e., 8 weeks), it is obvious that this was sufficient for adaptations of the cardiorespiratory system of our participants.

### 4.3. Differential Effects of Two Types of Endurance Exercise

Although both experimental groups made significant progress in endurance capacity, a superior improvement was evident in the TEE group. The authors are of the opinion that such differential effects are mainly the result of certain mechanisms: (i) the individualised training intensity exclusively applied in TEE, and (ii) the improvement of the running economy in TEE.

Since the response to physical interventions (including PE) depends on many influencing factors, individualisation is vital in maximising the efficiency of PE intervention [62]. Indeed, irrespective of the fact that both experimental groups exercised in the aerobic zone (i.e., bellow the anaerobic threshold), it cannot be ignored that strictly individualised workload could be practically applied only in the TEE group. Prior to the exercise protocols observed herein, the subjects were tested via the Conconi test to determine their anaerobic threshold. The test was repeated every two weeks to adjust the training load to the adaptations of the cardiorespiratory system. The use of heart rate chest belts and easy adjustment of the speed of movement and incline of the treadmill allowed the TEE participants to consistently exercise beneath the proper thresholds.

On the other hand, participants in the CAT group performed the PE based on dance choreography. Choreographed aerobic exercise (i.e., aerobic dance) is a highly motivational form of exercise, but the physiological demands of the exercise cannot be individualised with regard to intensity [63]. While participants in the CAT program logically differed in their fitness status, the intensity could not be adequately optimised for all participants, although the instructor modified the intensity on the basis of the feedback received by participants (please see the Methods for more details). However, during the CAT it is practically impossible to meet the demands for each single participant, which is known to be one of the most important factors for reaching appropriate training goals [64]. As a result, some participants did not exercise according to designated thresholds. Consequently, they could not optimally improve their endurance capacity, the possibility of which is already demonstrated for similar forms of aerobic dance exercises [65].

Another reason for the differential effect of the two types of PE is found in the measurement tool we used to assess endurance (Rockport walk/run test). It can be assumed that the TEE group improved the walking/running economy more than participants from the CAT group simply because TEE training mimics the testing protocol of the Rockport test. Logically, after 8 weeks of TEE, the participants familiarised themselves with walking/running on the treadmill, which almost certainly improved their walk/run economy and positively influenced their achievement on tests. Indeed, studies have confirmed that long-term endurance training improves the running economy in untrained and sedentary individuals [66]. This improvement certainly led to better performance on the Rockport test conducted via walking and/or running on the treadmill, consequently having superior effects in TEE. Again, the reliability of the Rockport test as a field testing procedure could be questioned, and because of that there is no doubt that the values of VO2 we reported should not be considered as absolutely relevant. However, as for previously discussed body composition, we believe that the measurement tool did could not greatly influence the fact that training groups improved their capacity to a greater extent than control group.

### 4.4. Strengths and Limitations

The most important limitation of the study is the fact that endurance capacity was not measured in a laboratory, but rather was estimated using a field testing procedure. Additionally, body composition (BF%) was estimated on the basis of skinfold measurement at only three sites (e.g., triceps, thigh, and suprailiac), while more reliable calculations include skinfold measurement on up to seven body sites. Moreover, participants were not randomly allocated into each group, but participated in each training program based on their own preferences. Finally, the study did not observe important motor variables (i.e., strength, flexibility) which are known to be important determinants of overall health status and are directly related to quality of life outcomes.

This is one of the few studies examining the effects of two different endurance-based exercise programs on indices of cardiovascular health in young women, and included a relatively large set of variables. We hope that the investigation herein will support further research in this area.

## 5. Conclusions

The results of this study confirmed the positive effects of both endurance exercise programs on anthropometry indices, the lipid panel, and endurance capacities. Positive changes occurred, although PE interventions were relatively short in duration (i.e., 8 weeks). Since the participants in this study were grouped based on their personal preferences concerning different forms of PE, we can conclude that this study design contributed to the effectiveness of the exercise.

The significant importance of the positive effects of exercise is highlighted by the fact that we observed young, healthy women. Naturally, they had a relatively low ceiling for improvement. Therefore, the obtained results indicate that the optimal exercise intensity and self-chosen type of PE may result in positive effects, regardless of health and fitness status. The differential effects of the observed endurance programs regarding improvements in endurance capacity could be related to the strictly individualised training in the TEE group and their familiarisation with treadmill equipment, which was used to assess endurance capacity.

Given the prevalence of obesity and cardiovascular diseases, our results provide clear practical guidelines for preventing such health problems. In future studies, the effects of the observed types of PE on other fitness and health parameters should be analysed. Considering the importance of motoric indices (i.e., strength, flexibility) in everyday functioning, future studies evidencing the effects of various forms of PE on these indices are warranted. Additionally, further studies where effects of dietary interventions (including the control of type of fat consumed) and exercise programs will be evaluated are warranted.

## Figures and Tables

**Figure 1 healthcare-09-00449-f001:**
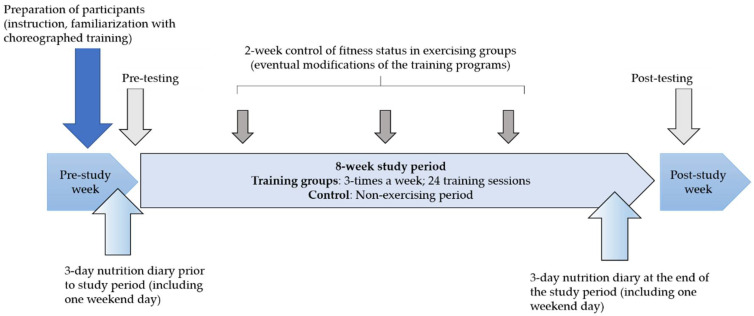
Design of the study.

**Figure 2 healthcare-09-00449-f002:**
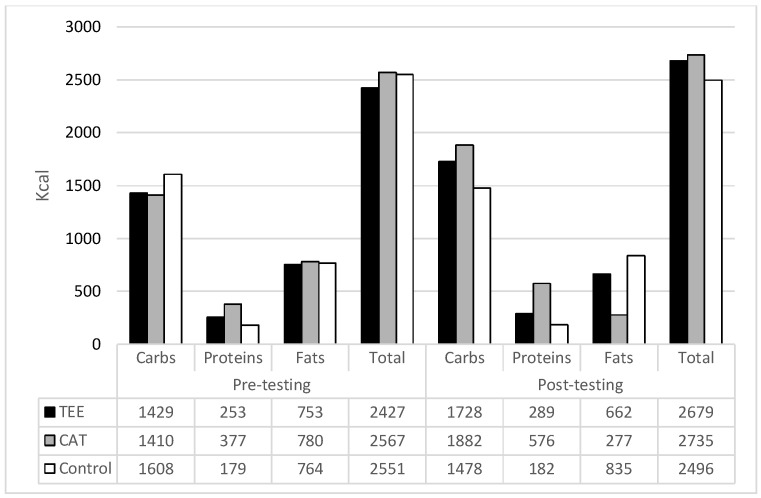
Caloric intake for the treadmill endurance exercise groups (TEE), choreographed aerobic training group (CAT), and control-group (C) at the beginning (Pre-), and at the end of the study (Post-).

**Table 1 healthcare-09-00449-t001:** Results of the analysis of variance for main effects (Group and Measurement), and interaction (Group x Measurement) with effect size values (η^2)^.

	Measurement			Group			Interaction		
	F Test	*p*	η^2^	F Test	*p*	η^2^	F test	*p*	η^2^
BM	426.1	0.001	0.88	0.26	0’.77	0.01	85.1	0.001	0.75
BMI	178.04	0.001	0.76	0.74	0.48	0.03	68.29	0.001	0.71
BF	112.1	0.001	0.66	2.43	0.09	0.08	28.93	0.001	0.5
END	534.52	0.001	0.9	5.41	0.01	0.16	128.39	0.001	0.81
PG	47.49	0.001	0.45	47.49	0.001	0.45	7.48	0.01	0.2
TC	137.05	0.001	0.7	1.45	0.23	0.04	25.5	0.001	0.47
HDL	151.39	0.001	0.73	3.36	0.03	0.12	41.09	0.001	0.59
LDL	205.59	0.001	0.78	2.63	0.08	0.08	31.11	0.001	0.52
TG	700.96	0.001	0.92	7.97	0.001	0.22	67.57	0.001	0.7

LEGEND: BM—body mass, BMI—body mass index, BF—body fat, END—Rockport endurance capacity test, PG—plasma glucose, TC—total cholesterol, HDL—high density lipoprotein, LDL—low density lipoprotein, TG—triglycerides.

**Table 2 healthcare-09-00449-t002:** Descriptive statistics and post hoc differences for study variables in each group (C—Control, CAT—choreographed aerobic training, TEE—treadmill-based endurance training).

	C (*n* = 19)		CAT (*n* = 19)		TEE (*n* = 20)	
	Pre-Test	Post-Test	Pre-Test	Post-Test	Pre-Test	Post-Test
BM (kg)	65.38 ± 8.32	64.79 ± 9.1 *	67.88 ± 5.86	59.91 ± 5.87 *	67.35 ± 8.39	59.38 ± 8.56 *
BMI (kg/m^2^)	23.77 ± 2.71	24.2 ± 2.49	24.45 ± 2.49	21.58 ± 2.32 *	25.11 ± 2.69	22.09 ± 2.79 *
BF (%)	33.94 ± 6.06	34.02 ± 6.22	34.77 ± 5.79	26.15 ± 5.41 *	36.43 ± 5.43	27.14 ± 4.04 *
END (mL/O2)	27.08 ± 2.26	28.17 ± 2.29 *	17.49 ± 1.53	29.76 ± 1.57 *	26.31 ± 1.58	32.62 ± 1.68 *^, C, CAT^
PG (mmol/L)	5.04 ± 0.55	4.94 ± 0.36	4.94 ± 0.48	4.52 ± 0.41 *	4.89 ± 0.54	4.25 ± 0.6 *^, C, CAT^
TC (mmol/L)	4.67 ± 0.57	4.52 ± 0.54	4.75 ± 0.47	4.03 ± 0.53 *	4.94 ± 0.56	3.77 ± 0.47 *^, C, CAT^
HDL (mmol/L)	1.6 ± 0.23	1.6 ± 0.26	1.63 ± 0.15	1.23 ± 0.16 *	1.74 ± 0.27	1.25 ± 0.18 *^, C, CAT^
LDL (mmol/L)	3.18 ± 0.6	2.99 ± 0.49	3.25 ± 0.47	2.44 ± 0.41 *^, C^	3.36 ± 0.43	2.21 ± 0.46 *^, C, CAT^
TG (mmol/L)	1.2 ± 0.22	1.02 ± 0.22 *	1.2 ± 0.17	0.57 ± 0.12 *^, C^	1.18 ± 0.23	0.72 ± 0.16 *^, C, CAT^

LEGEND: BH—body height, BM—body mass, BMI—body mass index, BF—body fat, END—Rockport endurance capacity test, PG—plasma glucose, TC—total cholesterol, HDL—high density lipoprotein, LDL—low density lipoprotein, TG—triglycerides, *—significant (*p* < 0.05) within group difference, ^c^ significantly (*p* < 0.05) different when compared to C, ^CAT^—significantly (*p* < 0.05) different when compared to CAT.

## Data Availability

Data will be provided to all interested parties upon reasonable request.

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
