# Peer review of "Indices of Cardiovascular Health, Body Composition and Aerobic Endurance in Young Women; Differential Effects of Two Endurance-Based Training Modalities"

_healthcare, 2021, doi:10.3390/healthcare9040449_

Round 1
Reviewer 1 Report
Abstract
The abstract is the first impact the readers have with the paper, so it is crucial to have it as clear and informative as possible. At present it should be improved especially in the first part, as it should be better explained why you come up with this aim based on a brief rational, trying to avoid focusing on the fact that only few studies examined this aspect.
Line 17: would it be better to include “apparently healthy”?
Line 20: I would suggest “during the experimental protocol” rather than “during the course of the study”
Introduction
This section should logically and subsequently lead the reader to the aim of the study. At present it is be improved to that it will be easier for the reader to clearly understand the development of the hypothesis leading to the methodology applied, for example including the rationale behind choosing to compare choreographed-based vs treadmill exercise. Maybe you might look to some literature referring to Zumba or other dance-based exercise.
Probably, some information regarding the dose-response could be also mentioned. Maybe you might want to have a look to these papers to better focus this section (https://doi.org/10.3390/jfmk5030048; https://doi.org/10.3390/jfmk5030063; https://doi.org/10.3390/jfmk6010031)
Methods
Line 92: as I mentioned for the abstract, make sure you can say “healthy”. I would suggest the use of apparently healthy here as well. This should be carefully considered, especially as you have overweight and other diseases (lines 106-108).
Figure 1 is really helpful. However, make sure you upload the one with the highest resolution possible. Also, you have “Conconi test” here but in the text you have the Rockport walking test for all the subjects. I think it will be better to include all the texts here.
Line 150-151. Any reference supporting this beside the authors’ experience?
As you are comparing two different modalities of exercise, how did you control for intensity? Did you match training for caloric expenditure? MET? Relative intensity? This is a crucial aspect of this study and should be clearly stated as you want to make sure that changes eventually occurring are due to modality, not on differences between intensity.
Figure 3 does not look professional. It should be improved.
Discussion
This section is quite detailed. However, it should be updated based on the comments to the previous sections.
Author Response
Abstract
The abstract is the first impact the readers have with the paper, so it is crucial to have it as clear and informative as possible. At present it should be improved especially in the first part, as it should be better explained why you come up with this aim based on a brief rational, trying to avoid focusing on the fact that only few studies examined this aspect.
RESPONSE: Thank you for your suggestion. The Abstract is rewritten, especially in the first part where more details on study background are added. Text now reads: “Endurance training (ET) has multiple beneficial effects on cardiovascular health, but there is an evident lack of knowledge on differential effects of various types of ET on indices of cardiovascular health in women. The aim of this study was to analyse the effectiveness of two different types of ET on changes in cardiovascular risk indicators in healthy adult women.” (please see beginning of the Introduction)
Line 17: would it be better to include “apparently healthy”?
RESPONSE: Thank you, amended accordingly.
Line 20: I would suggest “during the experimental protocol” rather than “during the course of the study”
RESPONSE: Thank you, amended accordingly.
Introduction
This section should logically and subsequently lead the reader to the aim of the study. At present it is be improved to that it will be easier for the reader to clearly understand the development of the hypothesis leading to the methodology applied, for example including the rationale behind choosing to compare choreographed-based vs treadmill exercise. Maybe you might look to some literature referring to Zumba or other dance-based exercise.
RESPONSE: Thank you, the Introduction is rewritten and now include more details about training effects of choreographed programs. Text reads: “For example, Barranco-Ruiz et al. studied sedentary females and evidenced im-provement in various cardiovascular risk variables as a result of 4-month long choreo-graphed endurance PE (e.g. Zumba fitness program) [22]. Cebula et al. applied 6-week Nordic walking endurance exercise and evidenced significant improvement in cardio-vascular function of sedentary postmenopausal women [23]. Chovanec and Gropel con-firmed the positive effects of 8-week endurance-based training on cardiovascular stress responses in untrained healthy female students [24]. The effects of combined-type training (45 min of aerobic exercise and 20min of resistance training) in women were also studied, and positive effects on morphological measures, blood pressure, and lipid panel were found [15]. Finally, a recent study showed that both resistance training and endurance training have a positive impact on the cardiovascular health indices of healthy young women, with no significant differences between groups conducting different forms of PE [16]. (please see 5th paragraph of the Introduction section)
Also, additional paragraph of the text is added in the Introduction where we explained the importance of lipid panel and anthropometrics in cardiovascular health (as suggested by 2nd Reviewer) – please see 4th paragraph of the Introduction
Probably, some information regarding the dose-response could be also mentioned. Maybe you might want to have a look to these papers to better focus this section (https://doi.org/10.3390/jfmk5030048; https://doi.org/10.3390/jfmk5030063; https://doi.org/10.3390/jfmk6010031
RESPONSE: Thank you for your suggestion. Two of three mentioned references are now included in the Discussion section.
Text reads: “Since the response to physical interventions (including PE) depends on many influencing factors, individualization is vital in maximizing the efficiency of PE intervention (Gronwald, Torpel, Herold, & Budde, 2020). Indeed, irrespective, etc. “ (please see Discussion; subsection 4.3)
and
“However, during the CAT it is practically impossible to meet the demands for each single participant, which is known to be one of the most important factors for reaching appropriate training goals (Foster et al., 2020).” (please see subsection 4.3.; 3rd paragraph)
Methods
Line 92: as I mentioned for the abstract, make sure you can say “healthy”. I would suggest the use of apparently healthy here as well. This should be carefully considered, especially as you have overweight and other diseases (lines 106-108).
RESPONSE: Thank you, amended accordingly.
Figure 1 is really helpful. However, make sure you upload the one with the highest resolution possible. Also, you have “Conconi test” here but in the text you have the Rockport walking test for all the subjects. I think it will be better to include all the texts here.
RESPONSE: Thank you for your suggestion. Figure is amended accordingly, and we included the better resolution of the image
Line 150-151. Any reference supporting this beside the authors’ experience?
RESPONSE: The reference supporting the statement is added, and text now reads: “This time frame for PE interventions was selected, as most drop-outs occur after two months of training (46)” (please see 1st paragraph of the subsection 2.3
As you are comparing two different modalities of exercise, how did you control for intensity? Did you match training for caloric expenditure? MET? Relative intensity? This is a crucial aspect of this study and should be clearly stated as you want to make sure that changes eventually occurring are due to modality, not on differences between intensity.
RESPONSE: Indeed, the intensity of the exercise was not sufficiently explained in the original version of the manuscript. In the revised version we tried to provide additional details about the intensity control in both training groups.
For the TEE text now reads: “In general, all participants performed all types of running protocols, but always at an intensity below the anaerobic threshold. The intensity of exercise in the TEE groups was controlled during each training session using the heart rate monitors, and participants were instructed to keep the heart rate between 60% and 80% of the maximal heart rate (approximately 120-155 beats/min, depending of the participant’s characteristics). The Conconi test was repeated every 2 weeks in order to adequately correct the training load.” (please see end of 2nd paragraph pf the subheading 2.3)
For the CAT group text reads: All participants, regardless of health and fitness status, performed the same protocol for the whole 8 weeks, which was controlled in real time by experienced aerobic dance in-structor who eventually increased/decreased the intensity of the exercise depending on perceived capacities of the group members. Specifically, at each training session three par-ticipants regularly wore heart rate monitors and controlled the intensity of exercise at each training session. They were instructed to report if their heart rate dropped below 120 beats/min, and/or raised about 160 beats/min). When two of three participants who wore a chest belt and controlled the training intensity reported the same discrepancy in heart rate (bellow or above the required frequencies), instructor modified the training program either by increasing the exercise demands (i.e. by changing the amplitude of movement, includ-ing/avoiding the jumps, increasing/decreasing the music tempo), which consequently re-sulted in increased/decreased intensity of the exercise.” (please see 3rd paragraph of the subsection 2.3)
Figure 3 does not look professional. It should be improved.
RESPONSE: The figure is amended accordingly, and now includes more details. Thank you.
Discussion
This section is quite detailed. However, it should be updated based on the comments to the previous sections.
RESPONSE: Thank you for recognizing the quality of our original discussion. In this version we made several amendments while incorporating yours and other reviewers’ suggestions (please see parts of the text in Discussion highlighted in yellow).
Staying at your disposal!
Authors
Reviewer 2 Report
Coronary risk, body composition and endurance capacity in young women; differential effects of two endurance-based training modalities
Abstract:
Adding the three groups of subjects together, the result is 59, not 58.
Introduction:
The title refers to coronary risk and the introduction refers to cardiovascular adaptations in general, without mentioning coronary risk. It is not possible to generalise or change terminology, this confuses the reader.
It is also necessary to add information on the variables related to coronary risk that will later be measured in the research.
Materials and methods:
How can you claim to do research with healthy subjects and then describe the sample by saying that 32% are overweight and 25% have high total cholesterol?
What methodology was used to obtain the skinfolds?
The calculation of anthropometric variables based on 3 skinfold measurements is less reliable than 5 or more. It would have been good to have taken more measurements.
Some of the measures used in the research (Rockport test, Conconi test, three skinfolds) are not very reliable. The use an unreliable measurement may be acceptable in some cases, but the use of many tests with a known major error generates doubts about the results obtained.
Claiming to compare the benefits of one activity to another while knowing the different intensities at which each activity is performed has no scientific relevance. The different adaptations associated with different training intensities are already known.
Results:
The results obtained are predictable and without any scientific relevance.
Although the results discuss the amount of macronutrients ingested by the participants, I have not found any reference to the control of the participants' diet during the study.
Although reference is made to the amount of macronutrients ingested, considering that HDL, LDL... are being assessed, I think it would be important to know/control the source of fats consumed during the study.
Author Response
REVIEWER 2
Abstract:
Adding the three groups of subjects together, the result is 59, not 58.
RESPONSE: Thank you for noticing it; this was typewriting mistake. Text now reads: “The sample included 58 young, healthy women (24±3 years; height: 165±6 cm, mass: 66.7±7.2 kg, BMI: 24.3±2.5 kg/m2, at baseline) divided in one control non-exercising group (n = 19), and two exercising experimental groups (EE). The first EE participated in choreographed aerobic-endurance training (CAT; n = 19), while the second participated in treadmill-based endurance exercise (TEE; n = 20) during the experimental protocol (8 weeks, 24 training sessions).”
Introduction:
The title refers to coronary risk and the introduction refers to cardiovascular adaptations in general, without mentioning coronary risk. It is not possible to generalise or change terminology, this confuses the reader.
RESPONSE: Indeed, the Introduction originally did not include sufficient details about coronary risk, and terms were used interchangeably. Therefore, in this version of the article we consistently used term “cardiovascular health” and/or “indices of cardiovascular health” (instead of cardiovascular risk). As a result, the title of the manuscript is changed and now reads: “Indices of cardiovascular health, body composition and aerobic endurance in young women; differential effects of two endurance-based training modalities
It is also necessary to add information on the variables related to coronary risk that will later be measured in the research.
RESPONSE: Thank you for this suggestion. In the amended Introduction we added one paragraph of the text explaining the variables of coronary risk we observed in the study. Text reads:
Among the many indicators of cardiovascular health, several variables deserve special at-tention, including variables of lipid panel, and anthropometric/body build status. Lipid panel (i.e., lipid test, lipid profile) is a set of variables describing the lipid levels in the blood, including total cholesterol (TC), high-density lipoprotein cholesterol (HDL), low-density lipoprotein cholesterol (LDL), and triglycerides (TG) [17]. The strategies to re-duce cardiovascular risk in primary and secondary prevention regularly focus on control (i.e. reduction) of lipid panel variables, particularly LDL [18]. Additionally, there is a glob-al consensus that poor cardiovascular health is strongly correlated with over-weight/obesity, with higher cardiovascular risk in overweight and obese individuals [8,19]. Therefore, evaluations of lipid panel, and anthropometric/body built status are valuable tools in assessing risk for cardiovascular problems [20,21]. Among other types of interventions, studies even have evaluated the effects of different types of PE on various indices of cardiovascular health status in women [8,16,22].” (please see end of 3rd and whole 4th paragraph of the Introduction)
Materials and methods:
How can you claim to do research with healthy subjects and then describe the sample by saying that 32% are overweight and 25% have high total cholesterol?
RESPONSE: Thank you for noticing it. Basically, subjects didn’t report and health-related problems during initial screening, but evidently cannot be considered as being healthy. Therefore, in this version of the manuscript we used the term “apparently healthy” in all circumstances. For example, text in the Abstract now reads: “The aim of this study was to analyse the effectiveness of two different types of ET on changes in indicators of CVH in apparently healthy adult women”. Also, in subsection on participants text reads: “Participants were 58 apparently healthy young women (24±2 years; height: 166±5.9 cm, mass: 66.9±7.6 kg, BMI: 24.4±2.7 kg/m2, at baseline).”, etc.
What methodology was used to obtain the skinfolds?
RESPONSE: Thank you for this question. The details on measurement are provided and text reads: “Anthropometric variables were measured according to the standard procedures of the International Society for the Advancement of Kinanthropometry [44]. A Seca stadiom-eter and scale (Seca, Birmingham, UK) and skinfold caliper (Holtain, London, UK) were used for measurements that occurred in the morning, before blood sampling. (please see subheading 2.2 – text highlighted in yellow)
The calculation of anthropometric variables based on 3 skinfold measurements is less reliable than 5 or more. It would have been good to have taken more measurements.
RESPONSE: We couldn’t agree more. However, you will also probably agree that the skinfold measurement is not “pleasant”, and therefore we tried to overcome this problem by measuring skinfold on “less poblematic” sites. Therefore, we used three-site equation (which includes Triceps, Thigh and Suprailiac skinfolds), and tried to avoid measurement of skinfold on other body regions (for example using the calculation of BF% based on skinfolds obtained at Chest, Axilla, Triceps, Subscapular, Abdominal, Suprailiac and Thigh). However, in this version of the manuscript this is clearly mentioned as one of the study limitations and text reads: “Also, body composition (BF%) was estimated on the basis of skinfold measurement at three sites (e.g. triceps, thigh, and suprailiac), while more reliable calculations include skinfold measurement on up to seven body sites” (please see subheading 4.4)
Some of the measures used in the research (Rockport test, Conconi test, three skinfolds) are not very reliable. The use an unreliable measurement may be acceptable in some cases, but the use of many tests with a known major error generates doubts about the results obtained.
RESPONSE: We agree that some of the tests used in study are not golden standards in measurement and evaluation; but we believe that this problem was at least partially solved by simultaneous usage of laboratory-tests (i.e. lipid panel), and approach in discussion. Also, in this version of the paper we tried to specify the problem of reliability of some tests in the discussion section. For example; text related to body composition (and measurement of skinfolds) reads: “One could argue that the changes we obtained can be partially a result of type of measurement, and questionable reliability of the method (i.e. we estimated BF% on the basis of skinfold measurement. However, the fact that all participants were measured by same evaluator, together with significant changes in training groups in comparison to non-significant changes of BF% in control group at least partially reduce the possibility that type of the measurement influenced our result.” (please see the end of 2nd paragraph in the subsection 4.2)
Also, for the endurance capacity measured by Rockport test text reads: “Again, the reliability of the Rockport test as a field testing procedure could be questioned; and because of that there is no doubt that the values of VO2 should not be interpreted. However, as for previously discussed body composition, we believe that the measurement tool did could not greatly influence the fact that training groups improved their capacity to a greater extent than control group.” (please see subheading 4.3, end of last paragraph)
Claiming to compare the benefits of one activity to another while knowing the different intensities at which each activity is performed has no scientific relevance. The different adaptations associated with different training intensities are already known.
RESPONSE: Thank you for this suggestion. Evidently, this issue deserved additional attention. Originally, we didn’t want to claim that “intensity was not controlled”, but rather that “intensity of the choreographed training was not controlled as accurately as it was the case in treadmill based training”. In the revised version of the manuscript we tried to be more specific and text is amended accordingly.
For the TEE text reads: “The intensity of exercise in the TEE groups was controlled during each training session using the heart rate monitors, and participants were instructed to keep the heart rate be-tween 60% and 80% of the maximal heart rate (approximately 120-160 beats/min, de-pending of the participant’s characteristics). The Conconi test was repeated every 2 weeks in order to adequately correct the training load.” (Please see 2nd paragraph of the subheading 2.3)
For the CAT text reads: “All participants, regardless of health and fitness status, performed the same protocol for the whole 8 weeks, which was controlled in real time by experienced aerobic dance instructor who eventually increased/decreased the intensity of the exercise depending on perceived capacities of the group members. Specifically, at each training session three participants regularly wore heart rate monitors and controlled the intensity of exercise at each training session. They were instructed to report if their heart rate dropped below 120 beats/min, and/or raised about 160 beats/min). When two of three participants who wore a chest belt and controlled the training intensity reported the same discrepancy in heart rate (bellow or above the required frequencies), instructor modified the training program either by increasing the exercise demands (i.e. by changing the amplitude of movement, adding/avoiding the jumps, increasing/decreasing the music tempo), which consequently resulted in increased/decreased intensity of the exercise. However, it cannot be said that the intensity was individualized because the characteristics of the group-program generally didn’t allow more accurate interventions in exercise intensity (Please see 3rd paragraph of the subheading 2.3).
Results:
The results obtained are predictable and without any scientific relevance.
RESPONSE:
We agree with your opinion that some results (i.e. absolute changes in studied variables) were somewhat predictable. At the end, previous studies confirmed positive effects of endurance based exercises in variables we observed. Because of that we initially hypothesized that the studied PE programs will have a positive influence on the studied variables (with no differential effects regarding PE type). However, we must mention that even the absolute changes we evidenced in this research were not always supported in previous literature. As you can see in some cases we had to explain several differences between our findings and findings obtained previously. For example, when explaining the effects on lipid profile one part of the text reads: “Slightly different results were reported by Polish researchers in a study of 35 young wom-en who were divided into underweight, normal weight, and overweight groups [43]. In this study, a significant improvement in the lipid profile was found only in the overweight group. The probable reason for the differences between the results of our study (e.g., posi-tive effects of PE in participants of normal weight) and the results from the previously cit-ed Polish study (e.g., no effect of PE in normal-weight participants) should be found in the differences of the PE programs applied in two studies. In particular, in our study, exercise intensity was individualised (please see Methods). This was not the case in the Polish study, and therefore, it is probable that their participants did not exercise at the desired and optimal heart-rate thresholds [43]. Also, with regard to anthropometric changes, difference between our results and results of previous study is discussed as it follows: “Specifically, Poelhman et al. stated that given the already low baseline levels, it was diffi-cult to reduce total or visceral fat in young women [55]. Given the similarity between par-ticipants (healthy young women in both studies), a possible explanation for the differ-ences between the obtained results (significant changes for anthropometric/body-build in-dices in our study vs. no changes in the investigation of Poelhman et al.) is provided next. …. Etc.” (please see parts of the text highlighted in green color in subsections 4.1 and 4.2).
On the other hand, we are not of the opinion that here evaluated differential effects of two studied exercise-programs (either evidenced, and those non-evidenced) could be expected and predicted; especially taking into account the lack of studies aimed to evaluate differential effects of different forms of endurance-based exercises. As you can see, our initial study hypothesis was only partially accepted (please see text highlighted in red in the 1st paragraph of the Discussion).
Although the results discuss the amount of macronutrients ingested by the participants, I have not found any reference to the control of the participants' diet during the study.
RESPONSE: It seems that we didn’t adequately highlight the fact that the study did not include the diet-modification, but rather “control” of the nutritional habits at the beginning, and at the end of the study. This included all three groups as presented in the Figure 2. We tried to specify it clearly in the revised version and text reads: “Although study did not include specific interventions in participants’ diet, nutrient intake, and caloric intake (in kcal) for all groups is evidenced before and at the end of the study period, and results is presented in Figure 2. In brief, no significant differences between groups were evidenced.” (please see end of the Results section; thank you)
Although reference is made to the amount of macronutrients ingested, considering that HDL, LDL... are being assessed, I think it would be important to know/control the source of fats consumed during the study.
RESPONSE: Thank you for this suggestion. We included this observation in the Conclusion section, and highlighted it as important issue worth studying in future, and text reads: “Also, further studies where effects of dietary interventions (including the control of type of fat consumed) and exercise programs will be evaluated are warranted. “ (please end of Conclusion section)
Thank you once again for your comments
Staying at your disposal
Reviewer 3 Report
Coronary risk, body composition and endurance capacity in young women; differential effects of two endurance-based training modalities.
The study examined the effects of two different endurance trainings (choreographed aerobic- exercise-CAT vs treadmill-based endurance exercise-TEE) on changes in cardiovascular risk indicators in healthy young women. Authors evaluated improvements both in terms of anthropometric/body composition indices, lipid panel and endurance capacity. In the end, they found that endurance exercise induced positive changes at all levels, with TEE improving endurance capacity to a greater extent than CAT.
The research article is quite well-designed as far as it includes a control group and two exercising groups. Exercise protocols are well described and structured. Statistical analysis is robust and the manuscript is quite good written.
Anyway I rise some concerns:
Major Revision:
Methods: authors define anthropometric variables for all recruited women. More information is needed about anthropometric features of each subgroups. Were the groups matched for age, height and BMI?
Methods: authors declare to let the participants choose the experimental protocol on their self-preferences. But no self-reported questionnaire was administered to subjects in order to evaluate their satisfaction about each training intervention. I guess this could have been a quite important issue, in order to evaluate the benefits on psychological well-being and the potential attendance in a long-term period.
Methods: Endurance capacity was not measured in laboratory but with field testing procedure. I suggest to modify the title as regard this aspect.
Minor Revision:
Introduction
Page 2, line 48: please reformulate the following sentence because it's too generic: “...while resistance training primarly affects the muscular system”
Author Response
REVIEWER 3
Coronary risk, body composition and endurance capacity in young women; differential effects of two endurance-based training modalities.
The study examined the effects of two different endurance trainings (choreographed aerobic- exercise-CAT vs treadmill-based endurance exercise-TEE) on changes in cardiovascular risk indicators in healthy young women. Authors evaluated improvements both in terms of anthropometric/body composition indices, lipid panel and endurance capacity. In the end, they found that endurance exercise induced positive changes at all levels, with TEE improving endurance capacity to a greater extent than CAT.
The research article is quite well-designed as far as it includes a control group and two exercising groups. Exercise protocols are well described and structured. Statistical analysis is robust and the manuscript is quite good written.
RESPONSE: Thank you for your support and for recognizing the potential in our study and manuscript. Also, we are particularly grateful for your suggestions. We tried to follow it and amended the manuscript accordingly.
Anyway I rise some concerns:
Major Revision:
Methods: authors define anthropometric variables for all recruited women. More information is needed about anthropometric features of each subgroups. Were the groups matched for age, height and BMI?
RESPONSE:
Thank you for noticing that we didn’t report sufficient details on study groups. Text is amended accordingly, and now reads: “The participants were offered several types of PE, and based on their self-preferences they were divided into two exercising experimental groups (EE) and one age-matched non-exercising group (n = 19; 23.8±2.0 years; height: 167±6.1 cm, mass: 65.4±8.3 kg, BMI: 23.8±2.7 kg/m2, at baseline). The first EE performed choreographed aerobic-endurance training (CAT; n = 19; 24±2 years; height: 166±6.0 cm, mass: 67.9±5.9 kg, BMI: 24.5±2.5 kg/m2, at baseline). The second EE participated in treadmill-based endurance exercise (TEE; n = 20: 25±2 years; height: 168.1±7.2 cm, mass: 67.4±8.4 kg, BMI: 25.1±2.7 kg/m2, at baseline).” (please see highlighted text – 1st paragraph of Methods)
Methods: authors declare to let the participants choose the experimental protocol on their self-preferences. But no self-reported questionnaire was administered to subjects in order to evaluate their satisfaction about each training intervention. I guess this could have been a quite important issue, in order to evaluate the benefits on psychological well-being and the potential attendance in a long-term period.
RESPONSE: We must agree that the problem of satisfaction is particularly important. In this study we didn’t screen the participants’ satisfaction with training program they participated, other than “drop-out” rates. However, drop-out rates are now specified in the Methods section and text reads: “Initially, the CAT group consisted of 23 participants, and 24 participants were involved in TEE program at the beginning. However, at the end of the study we included only those participants who participated at minimally 80% of training sessions, resulting in drop-out rate of17% (18% and 16% for CAT and TEE, respectively).”
Methods: Endurance capacity was not measured in laboratory but with field testing procedure. I suggest to modify the title as regard this aspect.
RESPONSE: Thank you, the title is amended accordingly and now reads: Indices of cardiovascular health, body composition and aerobic endurance in young women; differential effects of two endurance-based training modalities
Minor Revision:
Introduction
Page 2, line 48: please reformulate the following sentence because it's too generic: “...while resistance training primarly affects the muscular system”
RESPONSE: Thank you, the text is amended and now reads: “Specifically, endurance training develops cardiovascular and respiratory functions the most, while resistance training primarily affects strength and power, and other muscular capacities” (please see 2nd paragraph of the Introduction)
Thank you once again.
Staying at your disposal
Round 2
Reviewer 1 Report
I think the authors successfully dealt with the revision requests and the paper can be now accepted for publication.
Reviewer 2 Report
Thank you for clarifying the doubts raised. Congratulations for the work done.